# Lipid Deposition and Mobilisation in Atlantic Salmon Adipocytes

**DOI:** 10.3390/ijms21072332

**Published:** 2020-03-27

**Authors:** Marta Bou, Xinxia Wang, Marijana Todorčević, Tone-Kari Knutsdatter Østbye, Jacob Torgersen, Bente Ruyter

**Affiliations:** 1Nofima (Norwegian Institute of Food, Fisheries and Aquaculture Research), 1432 Ås, Norway; xinxiawang@zju.edu.cn (X.W.); marijana.todorcevic@ocdem.ox.ac.uk (M.T.); Tone-Kari.Ostbye@Nofima.no (T.-K.K.Ø.); jacob.seilo.torgersen@aquagen.no (J.T.); Bente.Ruyter@Nofima.no (B.R.); 2College of Animal Sciences, Zhejiang University, Key Laboratory of Animal Feed and Nutrition of Zhejiang Province, Hangzhou 310058, China; 3Oxford Centre for Diabetes, Endocrinology and Metabolism, Radcliffe Department of Medicine, University of Oxford, Oxford OX3 7LE, UK; 4AquaGen, PO Box 1240, N-7462 Trondheim, Norway

**Keywords:** lipogenesis, lipolysis, fasting, leptin, mitochondria, Salmo salar

## Abstract

The present study aimed to elucidate how Atlantic salmon adipocytes pre-enriched with palmitic (16:0, PA), oleic (18:1*n*−9, OA), or eicosapentaenoic (20:5*n*−3, EPA) acid respond to a fasting condition mimicked by nutrient deprivation and glucagon. All experimental groups were supplemented with radiolabeled PA to trace secreted lipids and distribution of radioactivity in different lipid classes. There was a higher content of intracellular lipid droplets in adipocytes pre-enriched with OA than in adipocytes pre-enriched with PA or EPA. In the EPA group, the radiolabeled PA was mainly esterified in phospholipids and triacylglycerols, whereas in the OA and PA groups, the radioactivity was mainly recovered in phospholipids and cholesterol-ester. By subjecting the experimental groups to nutrient-deprived media supplemented with glucagon, lipolysis occurred in all groups, although to a lower extent in the OA group. The lipids were mainly secreted as esterified lipids in triacylglycerols and phospholipids, indicating mobilization in lipoproteins. A significant proportion was secreted as free fatty acids and glycerol. Leptin secretion was reduced in all experimental groups in response to fasting, while the mitochondria area responded to changes in the energy supply and demand by increasing after 3 h of fasting. Overall, different lipid classes in adipocytes influenced their mobilization during fasting.

## 1. Introduction

Adipocytes are dynamically engaged in the regulation of whole-body energy homeostasis. In energy balance, lipolysis and re-esterification of free fatty acids (FA) are opposing processes taking place at the same time at similar rates in a futile cycle [1]. This allows adipocytes to quickly react and adjust to alterations in the state of energy balance by promoting the storage or mobilization of energy in case of a positive or negative energy balance, respectively. In fish, as in mammals, these processes are highly regulated by hormones, cytokines, and nutritional factors [2,3,4,5].

Lipolysis refers to the process by which triglycerides (TAG) are hydrolyzed to FAs and glycerol. These hydrolyzed FAs are primarily transported to other organs where they will be β-oxidized for energy production. A small part, however, can remain in the adipocyte where they can be either β-oxidized for energy purposes or re-esterified for further storage. The lipid oxidation capacity of Atlantic salmon (*Salmo salar*) adipose tissue is regulated by different FAs in vivo [6] and in vitro [5], being *n*-3 highly unsaturated fatty acids (HUFAs) able to promote FA oxidation rather than deposition. Several stimuli have been shown to elicit the lipolytic cascade in fish adipocytes. Thus, an increase in lipolysis was observed in adipocytes from rainbow trout (*Oncorhynchus mykiss*) [2] and gilthead seabream (*Sparus aurata*) [7] that fasted for three weeks or eleven days, respectively. An induction in lipolysis was also observed in adipocytes from gilthead seabream fed experimental diets containing plant protein [7] or vegetable oils [8]. Lipolysis is also under hormonal control in these species, where glucagon and growth hormone increase the lipolytic rate, whereas insulin decreases it [2,7].

Adipocytes have an important function as endocrine organs, being able to produce and secrete different hormones that can act both locally or peripherally [9,10]. In this sense, leptin is one of the most notable hormones due to its implication in regulating appetite, energy metabolism, growth, stress, and immune function across vertebrate groups [11]. In mammals, leptin is predominantly produced by adipocytes [12,13], and it is secreted in proportion to body adiposity [14], signaling the nutritional status. In fish, the plasma leptin source has not been determined yet. However, the liver has been suggested as the main producer due to its high *lep* expression [15,16,17]. Nevertheless, leptin has been reported to play an important role in regulating energy stores and their mobilization in rainbow trout [18].

In the present study, we evaluated adipocyte responses to three different FA: palmitic acid (16:0; PA), oleic acid (18:1*n*−9; OA), and eicosapentaenoic acid (20:5*n*−3; EPA). These FA were selected based on their relevance on current feeding practices. While the content of OA has increased in Atlantic salmon diets, that of PA and EPA has decreased [19,20]. Changes in dietary FA composition are reflected in the tissues and organs of the fish [21]. Even though these changes might not impact fish growth performance, they affect the nutritional quality of the fish by reducing the levels of the healthy FA, EPA, and DHA in their fillets [21]. Atlantic salmon has been ranked as the most efficient aquaculture production system [22,23], being an important source of protein, omega-3 FA, vitamins, and minerals for human consumption. Atlantic salmon are fed high-lipid diets in commercial production [19]. As in mammals, excess of energy is translated into increased abdominal fat deposition. The functions and development of Atlantic salmon white adipose tissue has many similarities with those from terrestrial vertebrates [5,6,24], which makes it a suitable and valuable experimental model. In this study, changes in adipocyte lipid dynamics were evaluated in terms of lipid droplet formation, cellular FA composition, and lipid classes composition. Additionally, we investigated the associations between these FA and adipocyte responses to an early stage of fasting. Adipocyte lipolysis, leptin production and secretion, and changes in mitochondrial area were measured.

## 2. Results

### 2.1. The Influence of OA, PA, and EPA on Lipid Droplets Formation, Total FA Content, and FA Composition

Differentiated adipocytes cultivated in media supplemented with 100 µM of OA, PA, or EPA for 72 h had, when observed by microscopy, different amounts of intracellular lipid droplets. The group supplemented with OA accumulated the highest number of large intracellular lipid droplets (Figure 1A,D), whereas cells supplemented with EPA contained less and smaller lipid droplets (Figure 1C,F), and few lipid droplets were observed in the cells supplemented with PA (Figure 1B,E). The microscopic observations agreed with the quantification of total FA by GC analyses, showing that the OA group contained a higher level of total cellular fatty acids than the EPA and PA groups (ANOVA; *p* = 0.0548) (Figure 1G).

The FA composition of the adipocytes was significantly affected by the FA supplementation to the culture media (Table 1). Thus, cells supplemented with OA for 72 h had a significantly higher content of OA (47%) than cells incubated with PA or EPA (16% and 18%, respectively; *p* = 0.002). Cells supplemented with PA for 72 h had a significantly higher content of PA (36%) than cells supplemented with OA and EPA (10% and 19%, respectively; *p* = 0.005). In a similar fashion, cells supplemented with EPA for 72 h had a significantly higher content of EPA (19%) than cells supplemented with PA (3%), whereas this FA was not detected in cells incubated with OA (*p* = 0.001).

### 2.2. The Influence of OA, PA, and EPA on Lipogenesis

In order to estimate the cellular lipogenic activity, the delta 9 desaturation capacity was analyzed by measuring the production of radiolabeled MUFA from radiolabeled PA. Despite of lack of significance, cells containing a high endogenous level of PA (PA group) presented a lower numerical ∆9 desaturation capacity than cells containing a lower level of this FA (OA and EPA groups) (ANOVA; *p* = 0.2197) (Figure 2A). Additionally, a significant reduction in the transcript abundance of the gene encoding the ∆9-desaturase enzyme (*scd*) was observed in the PA group (Figure 2B; *p* < 0.0001). Adipocytes from the EPA group, on the other hand, presented significantly higher *scd* mRNA levels than the other two groups (*p* < 0.0001).

### 2.3. The Influence of Endogenous FA Composition on Incorporation of Radiolabelled PA in Different Cellular Lipid Classes

To examine the effect of endogenous FA composition on the metabolism of [1-^14^C] PA, the relative incorporation of radiolabeled [1-^14^C] PA in the different intracellular lipid classes, such as phospholipids (PL), free fatty acids (FFA), triglycerides (TAG), diacylglycerols (DAG), and cholesteryl ester (CE), was analyzed by HPTLC (Table 2). In adipocytes enriched with PA or OA, nearly 50% of the radioactivity was found in PL, while more than 80% was recovered in PL in adipocytes enriched with EPA. Significantly more [1-^14^C] PA was found in FFA in the PA group (12%) than in the OA (2.6%) and EPA (3.4%) groups. Significant higher proportions of radiolabeled TAG were found in the EPA group (11.9%) than in the OA group (5.84%), whereas the PA group had intermediate values (11.3%). No difference was found in the proportions of [1-^14^C] PA recovered in DAG between the three treatments. More than 40% of the radioactivity recovered in cellular lipids was found in cholesteryl esters (CE) in the OA group, which was significantly higher than that in the PA group (27.6%) and in the EPA group (not detectable amounts). Overall, the total amount of nmols recovered in the cellular lipid fraction was 3.8 and 3.3 times higher in the EPA and OA group, respectively, than in the PA group.

### 2.4. Cellular Response to Nutrient Deprivation and Glucagon

#### 2.4.1. Leptin

The type of FA supplemented to the adipocytes contributed to leptin regulation at both transcript and protein levels, but the effect differed across the nutritional/physiological status of the cell. Leptin transcript levels were not significantly modified by the different FA tested (OA, PA, or EPA; *p* = 0.0873) (Figure 3A). However, serum deprivation together with glucagon supplementation for 18 h significantly reduced the transcript levels of leptin in all the experimental groups (*p* = 0.0004; Figure 3A). Measurements of the intracellular levels of leptin showed a significant interaction between the FA and culturing condition (*p* = 0.0065) in which leptin protein levels were reduced already after 3 h of serum deprivation and glucagon supplementation in adipocytes enriched with OA and maintained at low levels thereafter (Figure 3B). On the other hand, cells enriched with PA or EPA maintained their intracellular leptin levels across culture conditions. Under standard growth conditions (Control), adipocytes incubated with OA presented a significantly higher intracellular concentration of leptin than adipocytes incubated with EPA, whereas adipocytes incubated with PA had intermediate levels (Figure 3B).

Leptin secretion to the media was significantly affected by both the FA supplemented (*p* < 0.0001) and the culturing condition (*p* < 0.0001) (Figure 4). However, a significant interaction between these two factors was observed (*p* < 0.0001) in which the type of FA supplemented to the cells only influenced basal leptin secretion under standard growth conditions (Control). Thus, adipocytes incubated with PA secreted a significantly higher amount of leptin to the media, followed by adipocytes incubated with OA, whereas EPA incubation promoted a significantly lower leptin secretion (Figure 4). Serum withdrawn and an addition of glucagon for 3 h significantly reduced leptin secretion in all three experimental groups, being these levels maintained after 18 h (Figure 4).

#### 2.4.2. Transcriptional Responses to Serum Deprivation and Glucagon Supplementation

To evaluate the role of the different FA used and how they might influence the adipocyte response to a stimulus emulating a fasting condition, the transcript levels of key adipogenic and lipolytic markers were assessed. In this study, serum deprivation together with glucagon supplementation for 18 h triggered a tendency to increase the mRNA levels of hormone-sensitive lipase (*lipe*; *p* = 0.0629) (Figure 5A). Additionally, there was a significant main effect of the FA the adipocytes were supplemented with (*p* <0.0001), such that *lipe* mRNA levels were significantly lower in cells enriched with PA than in cells enriched with OA or EPA. The same was true for lipoprotein lipase (*lpl*) mRNA levels, though these differences provoked by the FA tested were restricted to standard culture conditions (*p* = 0.0017; Figure 5B). The overall tendency to increased mRNA levels of *lipe* and *lpl* in all three experimental groups indicates an increase in lipolysis when the cells are mimicking a fasted condition.

#### 2.4.3. Lipolysis

Cellular lipid release was evaluated by measuring radiolabeled-secreted lipid products from [1-^14^C] PA and total non-radiolabeled non-esterified free fatty acids (NEFA) and glycerol in the culture media after 3 and 18 h of serum deprivation and glucagon addition (Figure 6). Analysis evaluating the secretion of total glycerol (Figure 6A) and NEFA (Figure 6B) showed no significant changes with increasing the time of adipocyte exposure to serum deprivation and glucagon supplementation (*p* = 0.5637 and 0.1608, respectively). Nevertheless, there was a significant main effect of the FA used during standard culture conditions on the amount of NEFA secreted (*p* = 0.0424). Thus, adipocytes incubated with EPA secreted a higher amount of NEFA than adipocytes incubated with OA, whereas adipocytes incubated with PA had intermediate levels. No significant effect of the FA used during standard culture conditions on the amount of glycerol secreted was observed (*p* = 0.4685). However, longer exposure to serum deprivation together with glucagon supplementation significantly increased the secretion of radiolabeled lipids to the media (*p* < 0.0001; Figure 6C). The type of FA added to the adipocytes during standard culture conditions had no effect on the total amount of radiolabeled secreted lipids secreted after 3 h of serum deprivation and glucagon addition. However, there was a significant interaction between the FA used and cell culture conditions (*p* = 0.0112; Figure 6C) in which, after 18 h of serum deprivation and glucagon addition, adipocytes pre-enriched with PA or EPA secreted a significantly higher amount of lipids to the media than those pre-enriched with OA. Most of the radiolabeled lipids secreted were esterified in DAG+TAG and PL, with relatively little recovered in the form of FFA (Figure 7A–C). Adipocytes pre-enriched with PA or EPA also released small amounts in the form of CE/wax esters (WE), whereas not detectable amounts were observed in adipocytes pre-enriched with OA (Figure 7D).

#### 2.4.4. Mitochondria

The mitochondria area was significantly affected by the FA the adipocytes were supplemented with (*p* = 0.0001) and by the culture conditions (*p* <0.0001) (Figure 8A). After 3 h of serum deprivation and glucagon addition, the mitochondria area was increased. However, after 18 h of serum deprivation and glucagon addition, these values were similar to those found in adipocytes growth under standard conditions (Control). Regarding the effect of the FA used, the mitochondria area was higher in adipocytes incubated with OA or EPA than in adipocytes incubated with PA.

The changes in mRNA levels of mitofusin 1 (*mfn1*) and mitochondrial fission 1 (*fis1*), two genes encoding for proteins responsible for mediating mitochondrial fusion and fission, respectively, were assessed. The mRNA levels of *mfn1* were significantly affected by the FA used and the culture condition (*p* = 0.0224 and <0.0001, respectively; Figure 8B). Thus, serum deprivation and glucagon addition for 18 h significantly increased the mRNA levels of *mfn1*, being this increase more pronounced in cells incubated with EPA than in cells incubated with PA. In a similar fashion, the mRNA levels of *fis1* were a significantly affected by the FA used and the culture condition (*p* <0.0001 and 0.0030, respectively; Figure 8C). In this case, serum deprivation and glucagon addition for 18 h only increased the mRNA levels of *fis1* in adipocytes incubated with PA or EPA. Overall, PA had a higher capacity to increase the mRNA levels of this gene.

## 3. Discussion

### 3.1. Adipocyte Response to OA, PA, and EPA Supplementation during a Fed Status

The supplementation of OA to mature Atlantic salmon adipocytes lead to a higher production of intracellular lipid droplets and higher level of cellular FA than supplementation of EPA and PA. This is consistent with a previous in vitro study done in Atlantic salmon adipocytes, where OA lead to a higher degree of lipid filling and fat cell differentiation than EPA [5]. Additionally, an in vivo study carried out by the same authors showed that increased dietary levels of *n*−3 HUFAs resulted in lower fat percentage in white adipose tissue [6]. In mammals, the lipid-lowering effects of EPA are mediated by an induction of mitochondrial β-oxidation, both in hepatocytes [25,26] and in adipocytes [27]. All these results suggest that OA has the capacity to promote Atlantic salmon adiposity to a higher degree than EPA and PA.

The lower number of lipid droplets and total lipids in the PA group, and also the reduced uptake of the [1-^14^C] PA than in the EPA and OA groups, might be a result of a strict control of the intracellular content of saturated fatty acids (SFA) in order to prevent potentially toxic effects of an excessive accumulation of intracellular SFA. In human adipocytes, the coordinated upregulation of the elongation of FA and their desaturation by the Δ−9 desaturase enzyme (SCD) protects against SFA cell injury [28]. In the present study, most of the [1-^14^C] PA incorporated into the adipocytes in the three experimental groups was converted to the MUFA 16:1 and 18:1, presumably by SCD activity. The mRNA transcript abundance of *scd* measured in the three experimental groups correlated with the amount of [1-^14^C] PA incorporated into the cells. Thus, *scd* transcripts were significantly lower in adipocytes from the PA group than in the EPA and OA groups.

The present study showed that EPA enrichment of cells totally inhibited the formation of [1-^14^C] PA-cholesterol ester compared to cells enriched with OA and PA, where approximately 40% and 30% of [1-^14^C] PA, respectively, was esterified with cholesterol. However, EPA did not reduce the esterification into PL. This is in agreement with mammalian studies, showing that EPA decreases cholesterol esterification in rat hepatocytes by interfering with the transfer of activated fatty acids to cholesterol by acyl-CoA:cholesterol acyltransferase [29]. In another study, it was shown that the inhibition rates of cholesterol esterification in lipoproteins were highest by EPA, next by linoleic acid and PA, and lowest by OA [30]. Furthermore, polyunsaturated FA (PUFA) suppressed lecithin:cholesterol acyltransferase (LCAT) activities much stronger than SFA at physiological concentrations [30]. Our study therefore indicates that EPA has a similar inhibitory effect on cholesterol esterification in Atlantic salmon, as previously shown in mammals. However, whether this is due to effects on LCAT activity in Atlantic salmon remains to be elucidated. A recent study revealed that PA esterification in CE, but not in PL, was directly correlated with body fat storage [31]. The higher incorporation of [1-^14^C] PA in the CE fraction observed in the OA group in our study also coincided with the highest total cellular lipid content.

### 3.2. Adipocyte Response to Mimicking a Fasting Condition: Lipid Mobilization

Scarce information is available regarding the mechanisms governing lipolysis in fish adipocytes. In the present study, Atlantic salmon adipocytes responded to an emulated fasting condition by secreting glycerol and NEFA to the media already after only 3 h of stimulation. This is in agreement with previous reports from other fish species [2,7]. However, the levels of these products in the media were maintained after 18 h of fasting stimulation. While the levels of glycerol released to the media were similar in all the experimental conditions, those of NEFA were affected by the FA adipocytes were enriched with, being adipocytes cultivated with EPA the ones secreting a higher amount of NEFA. A similar response has been previously reported in rat adipocytes, where SFA reduced lipolytic activities when compared to PUFA [32,33]. Interestingly, there was a marked increase in secreted radiolabeled lipids in response to extending fasting from 3 to 18 h, being the secretion higher in the EPA and PA groups than in the OA group. This was probably due to the fact that, in the OA group, [1-^14^C] PA was primarily esterified in CE and PL and very little in TAG, while more [1-^14^C] PA was esterified in TAG in the EPA and PA groups. This is in agreement with studies showing that, during lipolysis, FA stored as TAG in adipocytes from different mammalian models are selectively mobilized according to molecular structure regardless of their content [34,35]. In our study, both the secretion of FA and radiolabeled products were higher in the groups with the lower level of intracellular lipid droplets but with the highest level of intracellular [1-^14^C] PA-TAG. Conversely, the lowest secretion of these products was observed in the OA group, which presented the highest level of [1-^14^C] PA esterified in CE. These results are in agreement with a study from mammals showing that EPA membrane incorporation reduces the cholesteryl ester mobilization from lipid droplets [36].

Interestingly, most of the secreted lipids were esterified in TAG and PL, with relatively little recovered in the form of FFA in all experimental groups. This mobilization of lipids, where lipolysis does not seem to be governed by secretion of glycerol and NEFA, might indicate that fish adipocytes are able to secrete proteins that play a key role mediating lipid transport. In this sense, apolipoprotein E is highly expressed in adipocytes, both in mammals [37] and in Atlantic salmon [24]. A mouse study showed that, in addition to apolipoprotein A1, apolipoprotein E may promote de novo biogenesis of HDL [38]. Further mechanistic studies are needed in order to better understand fish lipolysis and transport pathways, as well as the possible role that secreted lipids might have as signaling molecules.

Fasting did not modify the mRNA levels of *lipe* nor *lpl*, two enzymes controlling the turnover of FA in adipose tissue. Nevertheless, a lack of correlation between cellular mRNA levels and activities of these lipases has been described and associated with extensive posttranscriptional and posttranslational regulation processes [39,40].

### 3.3. Leptin Regulation in Adipocytes Enriched with OA, PA, or EPA during a Fed and a Fasting Status

Despite of a lack of regulation observed at a transcriptional level, the three FA tested in the present study triggered different adipocyte responses in terms of intracellular and secreted leptin. Thus, adipocytes enriched with EPA presented the lowest intracellular and secreted levels of leptin, whereas cells enriched with PA had high intracellular levels of leptin and the greatest leptin secretion capacity. In mammals, leptin levels reflect body lipid content acting as a negative feedback adipostatic signal to control energy homeostasis [41,42]. Interestingly, in the present study, adipocytes enriched with PA secreted higher amounts of leptin regardless of not presenting the highest lipid content. Recent studies in rainbow trout point to the contribution of visceral adipocytes to plasma leptin levels [18,43]. Additionally, it has been suggested that leptin secretion in salmonids is not correlated to the size of the secreting tissue [18], being rather differentially regulated depending on the environmental and physiological conditions. The present results indicate that individual fatty acids have the capacity to differentially regulate leptin production and secretion. This observation is in line with research done in humans, where plasma leptin concentrations were influenced by the dietary type of fat [44]. In this sense, dietary *n*−3 PUFA has been reported to reduce human leptin transcript abundance both in vivo and in vitro and to negatively correlate with plasma leptin concentrations [45].

When a fasting state was mimicked, the transcript abundance of leptin was decreased, and the intracellular protein leptin levels were significantly reduced in adipocytes pre-enriched with OA. Additionally, a consistent and significant decrease in leptin secretion in response to fasting was observed in all experimental groups. This response agrees with that from most mammalian species [42,46,47,48], where fasting acutely reduces serum leptin. Several studies evaluating the effects of fasting or feed restriction in teleost fishes have reported increases in leptin synthesis and secretion (reviewed by [11]). However, leptin responses induced by fasting are diverse, even within the same fish species. While some studies have reported an increase of plasma leptin levels in rainbow trout [49] and Atlantic salmon [50] subjected to fasting or to a restricted diet, others have reported no effect in these two species [15,18]. All these discrepancies might be related to differences in experimental settings (laboratory conditions vs. ambient conditions), protocols (duration of the imposed fasting), differences in sample size, differences in fish life stages, and gender differences, among others. Additionally, due to genome duplication events, Atlantic salmon possess two leptin paralog-pairs (LepA1/LepA2 and LepB1/LepB2) [15,51] and two paralog receptors (LepRA1/LepRA2) [52]. The existence of leptin paralogs with possible functional diversification in a species or in a tissue dependent manner further complicates the picture.

### 3.4. Mitochondrial Responses in Adipocytes Enriched with OA, PA, or EPA during a Fed and a Fasting Status

Mitochondria are highly dynamic organelles that change morphology depending on the cellular context by means of coordinated fusion and fission events [53,54,55]. In the present study, the mitochondrial area was increased after 3 h of nutrient deprivation and glucagon stimulation, being this increase particularly noticeable in adipocytes pre-enriched with OA or EPA. These results are consistent with previously reported observations relating changes in mitochondrial architecture with the balance between energy supply and demand [56]. Fasting is known to induce a metabolic response by which energy production from mitochondria is elevated in order to ensure energy supply. However, this elevates the risk of mitochondrial oxidative damage [57]. In general, mitochondria in cells exposed to a fasting condition tend to elongate and interconnect shortly after nutrient depletion [57,58,59,60]. It has been reported that mitochondrial fusion protects metabolically challenged mitochondria by reducing reactive oxygen species upon fasting [57]. Interestingly, after 18 h of mimicking a fasting status in the present study, the mitochondrial area was similar to that present in the control cells (“fed cells”). The observed arrest in fusion and increase in mitochondria fragmentation might indicate the first steps towards nutrient-induced apoptosis. However, further studies are needed in order to unravel the physiological role of mitochondrial dynamics in fish adipocytes.

The fact that the extent of mitochondrial fusion differed between adipocytes pre-enriched with different FA indicates that this process can be modulated according to single metabolites. This has been shown to be the case in humans, where ingestion of the FA stearic acid (18:0) caused mitochondrial fusion within 3 h after ingestion, whereas this response was not observed after PA ingestion [61]. Our results showed that OA and EPA had a greater capacity than PA to promote mitochondrial dynamics in Atlantic salmon adipocytes. In this sense, hepatocytes from Atlantic salmon-fed diets containing high levels of EPA [62], as well as salmon hepatocytes with high levels of exogenously added EPA [63], presented an increased mitochondrial area. However, a significantly smaller mitochondrial area was observed in hepatocytes from Atlantic salmon fed a rapeseed oil diet rich in OA when compared to those fed the EPA diet [62]. This might indicate that these responses do not only depend on the specific FA but also on the cell type.

## 4. Materials and Methods

### 4.1. Isolation of Preadipocytes

Atlantic salmon with a mean body weight of 3.46 (SEM 0.58) kg and a mean body length of 617 (SEM 3) cm were obtained from Nofima Research station at Averøy, Norway. Fifteen fish were randomly selected and transported from sea cages to land tanks with recirculating water. One at the time, the fish were anaesthetized with metacain (MS-222; 0.08 g/L) and killed by a sharp blow to the head. The arch bows of the gills were cut, and after bleeding for a few minutes, the abdomen was cut open to expose the visible white adipose tissue surrounding the intestinal tract. The experiment was conducted according to the National Guidelines for Animal Care and Welfare of the Norwegian Ministry of Research (FOR-2015-06-18-761) and classified as not requiring a specific license (§2-f, corresponding to Directive 2010/63/EU Article 1, section 5f), since the experimental treatments were not expected to cause any distress or discomfort for the fish, being that the fish was dead prior to adipose tissue dissection. Visceral adipose tissue was excised, and Atlantic salmon preadipocytes were isolated and cultured as described by Todorčević et al. [5]. Briefly, the dissected adipose tissue from 15 fish was washed with Leibowitz-15 (L-15) and then minced into small pieces. After washed twice with L-15, the tissue was digested in 0.2% collagenase in L-15 (1 g tissue / 5 mL L-15) at 13 °C for 1 h under shaking and then filtered through 250 and 100 μm nylon filters to remove large particulate materials. The resulting suspension was then centrifuged at 500× *g* for 10 min at 10 °C. The buoyant lipid layer with mature adipocytes and the digestion medium was removed immediately by aspiration. After washing twice, the sedimented cells were resuspended in a growth medium containing L-15; 10% fetal bovine serum (FBS); 2mM L-glutamine; 10mM HEPES; and 1% antibiotics (a mixture of penicillin, streptomycin, and amphotericin B) and seeded onto laminin-coated 6- and 24-well plates at a density of approximately 10 g tissue/25 cm^2^. The cells were incubated at 13 °C, and most cells were attached the next day. The medium was changed every 3 days until the cells reached confluence after approximately 1 week. Confluent cells were cultivated for 48 h in differentiation-inducing medium containing growth medium supplemented with 0.5 μM dexamethasone, 5 nM triiodothyronine, 12 μM isobutylmethylxanthine, and 10 μg/mL insulin.

### 4.2. Incubation of Adipocytes with Radiolabelled Palmitic Acid

After being incubated in differentiation medium for 2 days, the cells were divided into 3 treatment groups and incubated with radiolabeled palmitic acid (PA) and one of the three different non-radiolabeled fatty acids (FA) tested. Thus, the cells were incubated with either 10µM [1-^14^C] PA (0.5 µCi/mL) + 100 µM PA (PA group), 10 µM [1-^14^C] PA + 100 µM OA (OA group), or 10 µM [1-^14^C] PA + 100 µM EPA (EPA group) in growth medium containing 2% FBS. Each incubation was conducted in six independent parallel experiments (*n* = 6), each representing adipocytes coming from a pool of adipose tissue from an average of 5 fish. The radiolabeled and non-radiolabeled FA were added to the medium in the form of their potassium salts bound to FA free bovine serum albumin (BSA) (the molar ratio of FA to BSA was 2.7:1). The radiolabeled PA was obtained from American Radiolabeled Chemicals, Inc. (St. Louis, MO, USA), and the non-radiolabeled FAs were all supplied from Sigma-Aldrich (St. Louis, MO, USA). The doses were selected based on previous in vitro studies performed on adipocytes [64,65] and based on extensive experience in our laboratory from previous studies.

Prior to incubation, aliquots of 10, 20, 30, 40, and 50 µL of the incubation medium with [1-^14^C] PA were transferred into vials containing 5 mL of liquid scintillation fluid (InstaGel II Plus; Packard Instrument, Downers Grove, IL, USA), and the total radioactivity was measured in a Tri-Carb 1900TR Liquid Scintillation Analyzer (Packard Instrument Company, Meriden, CT, USA). The specific radioactivity (cpm/nmol FA) was subsequently calculated for PA substrate.

The cells were incubated with radiolabeled PA and non-radiolabeled FA for 72 h at 13 °C. After incubation, the culture medium was used for determination of total radioactivity. Three replicates from each treatment group, each representing adipocytes coming from a pool of adipose tissue from an average of 5 fish, were washed twice in PBS that contained 1% albumin, washed once more with regular PBS, and harvested in 500 µL of PBS and stored at −40 °C prior to the analysis of radiolabeled lipid classes. The other 3 replicates from each treatment group were washed twice in PBS that contained 1% fatty acid free albumin, washed once more with regular PBS, and incubated with serum-free growth medium containing glucagon (5 µg/mL) for 3 and 18 h. After the 3 h incubation, 1 mL medium from each cell well was collected and stored at −40 °C before the secreted radiolabeled lipid classes, leptin, non-esterified free fatty acids (NEFA), and glycerol were analyzed (3 h GLU). The cells went on the treatment until 18 h (18h GLU), and then, the cells were harvested and the medium was collected for the same analyses described above.

### 4.3. Incubation of Adipocytes with Non-Radiolabelled Fatty Acids

Differentiated adipocytes were washed with serum-free growth medium and treated with incubation medium, which consisted of growth medium with 2% FBS and one of the three treatments consisting of: 10 µM PA+100 µM PA (PA group), 10 µM PA+100 µM OA (OA group), or 10 µM PA +100 µM EPA (EPA group). The incubation continued for 72 h at 13 °C, where each group contained 8 replicates, each representing adipocytes coming from a pool of adipose tissue from an average of 5 fish. After the incubation time, 4 replicates from each treatment group were washed twice in PBS that contained 1% albumin, washed once more with regular PBS, harvested in RLT buffer containing β-mercaptorthanol, and stored at −80 °C prior to RNA extraction. This time point is referred to as the Control. The other 4 replicates from each treatment group were washed 3 times in PBS that contained 1% albumin, washed once more with regular PBS, and incubated with serum-free medium containing glucagon (5 µg/mL) for 18h. After incubation, the cells were washed and harvested for gene expression analysis, as described above. This time point is referred to as 18 h GLU.

### 4.4. Leptin, NEFA, and Glycerol Secretion Measurements

Leptin secretion to the culture medium at the different time points was determined by a commercial kit (Cusabio Ltd., Wuhan, China) specific for Atlantic salmon. The method was based on a competitive inhibition enzyme immunoassay technique. A competitive inhibition reaction is started between leptin (standards or samples) and HRP-conjugated leptin with the pre-coated antibody specific for Atlantic salmon leptin. The measurements were conducted according to the protocol supplied by the kit.

NEFA concentration in the medium at the different time points was determined by an enzymatic colorimetric method using a commercial kit (Randox Laboratories Ltd., Co. Antrim, UK). The principle for the test involves acylation of coenzyme A by FA in the sample in the presence of acyl-CoA synthetase and production of hydrogen peroxide in the presence of acyl-CoA oxidase. Color was measured at 550 nm in a Wallac 1420 VICTOR3TM Multilabel counter spectrophotometer (PerkinElmer, Wellesley, MA, USA).

Glycerol concentration in the medium was examined by a glycerol cell-based assay kit (Cayman Chemical Company, Ann Arbor, Michigan, USA). Glycerol present in the sample is phosphorylated by adenosine triphosphate (ATP), forming glycerol-1-phosphate (G-1-P), and adenosine-5′-diphosphate (ADP) in the reaction catalyzed by glycerol kinase. G-1-P is then oxidized by glycerol phosphate oxidase to dihydroxyacetone phosphate (DAP) and hydrogen peroxide (H_2_O_2_). A quinoneimine dye is produced by the peroxidase-catalyzed coupling of 4-aminoantipyrine (4-AAP) and sodium *N*-ethytl-N-(3-sulfopropyl)m-anisidine (ESPA) with H_2_O_2_, which shows an absorbance maximum at 540nm. The increase in absorbance at 540nm, measured in a Wallac 1420 VICTOR3TM multilabel counter spectrophotometer, is directly proportional to the glycerol concentration in the sample.

### 4.5. Lipid Extraction and Radiolabelled Lipid Class Analysis

Total lipids from adipocytes incubated with the different FA (PA, OA, or EPA) were extracted as described by Folch et al. [66]. The chloroform phase (300 µL) with butylated hydroxytoluene (BHT) (final concentration 0.7 mg/L) was divided into two parts. One part (150 µL) was dried under nitrogen gas, and the residual lipid extract was redissolved in 20 µL of chloroform. Phospholipid (PL), free fatty acids (FFA), diacylglycerol (DAG), triglycerol (TAG), and cholesterol ester (CE) were separated by thin-layer chromatography (TLC) using a mixture of petroleum ether, diethyl ether, and acetic acid (131:20:1, v/v/v) as the mobile phase. The chloroform phase was applied onto the TLC-plate (Merk HPTLC Silica gel 60 F254, 10 × 20 cm) using a Linomat 5 (Camag, Switzerland) and dried. The plates were kept in a chamber with a saturated atmosphere of the mobile phase until the liquid reached 2 cm from the upper edge of the plates. The spots corresponding to FFAs, PLs, DAGs, TAGs, and CEs were identified by comparison with known standards by a Bioscan AR-2000 Radio-TLC & Imaging Scanner and quantified with the WinScan Application Version 3.12 (Bioscan Inc., Washington, DC, USA).

The rest of the chloroform phase (150 µL) was used to evaluate the degrees of unsaturation and elongation by silver-ion TLC as described by Nikolova-Damyanova [67]. The chloroform phase was dried under nitrogen gas, and the residual lipid extract was transmethylated overnight with, 2 mL benzene, 2 mL methanolic HCl, and 200 µL 2’,2’-dimethoxypropane at room temperature. Two milliliters of hexane was added, and the samples were neutralized with NaHCO_3_. The benzene/hexane phase, which contained methylated lipids, was collected, dried at 60 °C under nitrogen gas, and redissolved in 10 µL chloroform. The lipids were separated on silica gel plates impregnated with silver nitrate (2 g silver nitrate in 20 mL acetonitrile) in toluene/hexane (40:60, *v*/*v*), and specific FAs were identified by comparison with known standards by a Bioscan AR-2000 Radio-TLC & Imaging Scanner (Bioscan Inc., Washington, DC, USA). The peaks corresponding to 16:0; 16:1, *n*−7; and 18:1, *n*−9 were scraped off into vials and dissolved with 5 mL of scintillation fluid and measured in a TRI-CARB 1900 TR scintillation counter.

### 4.6. Immunofluorescence Staining

Cells seeded in 24-well plates with a type 1.5 cover slip glass bottom were used for investigating the cellular localizations of leptin and mitochondria in (I) differentiated adipocytes incubated with non-radiolabeled FA for 72 h (Control), (II) differentiated adipocytes incubated with non-radiolabeled FA for 72 h followed by a 3-h incubation with glucagon (5 µg/mL), and (III) differentiated adipocytes incubated with non-radiolabeled FA for 72 h followed by an 18-h incubation with glucagon (5 µg/mL), as described in Section 2.3. After the different incubation conditions, cells were washed twice with PBS containing 1% albumin and fixed with 1% paraformaldehyde in PBS for 10 min at room temperature. Visualization of mitochondria was achieved with MitoTracker Red CMXRos (Life Technologies). Immunofluorescence was carried out on saponin permeabilized cells (0.2% saponin in 1 × PBS). Immunofluorescence analysis of adipocyte leptin (1:100) (PMID:21775646, R&D Systems, Inc., Minneapolis, MN, USA) was carried out for all treatments and incubation times (Control, 3h GLU, and 18h GLU). Images were captured and analyzed using Zeiss Axiovision Z1 and Zeiss Axiovision software, respectively (Carl Zeiss Microimaging GmbH, Göttingen, Germany). Additionally, the adipocytes were imaged using contrast microscopy (DIC) to visualize the lipid droplet number and spatial distribution within the cells.

### 4.7. RNA Extraction, cDNA Synthesis, and Quantitative Real-Time RT-PCR

Total RNA was extracted by using an RNeasy^®^ mini kit, according to the manufacture′s instructions. RNA was treated with RNAase-free DNase I to remove any contaminating DNA. Approximately 350 ng of total RNA was reverse-transcribed into cDNA using a TaqMan^®^ Gold RT-PCR Kit (Applied Biosystems, Foster City, CA, USA) in a 20 µL reaction system according to the manufacturer′s protocol. The PCR primers (Table 3) were designed using the Vector NTI (Invitrogen, Carlsbad, CA, USA) and synthesized by Invitrogen. Efficiency was checked from two-fold serial dilutions of cDNA for each primer pair. Real-time PCR was performed in duplicates in 96-well optical plates on a Light Cycler 480 Instrument (Roche Diagnostics, Mannheim, Germany) with gene-specific primers. Three commonly used reference genes (β-actin, elongation factor 1A, and eukaryotic translation initiation factor 3) were tested for stability using GeNorm and NormFinder. Finally, β-actin met the criteria of stability in the analyzed material. PCR master mixes consisted of l μL forward and reverse primers (final concentrations of 0.5 μM), 4 μL 1:10 dilution of cDNA, and 5 μL Light Cycler 480 SYBR Green I Master (Roche Applied Science, Mannheim Germany). The specificity of PCR amplification was confirmed by a melting curve analysis. Relative quantification of the abundance of transcripts was calculated using the ΔΔCT method and formula −ΔΔCT = − [(Ct_target gene_ −Ct_β-actin_)_treatment_ − (Ct_target gene_ − Ct_β-actin_)_control_] [68].

### 4.8. Protein Measurements

Protein concentrations were assayed using a total protein kit (Sigma, St. Louis, MO, USA) based on the method of Lowry [69] and modified by Peterson [70]. Standards were prepared by diluting 400 μg/mL of BSA in water. Sodium chloride (final concentration of 0.1 M) was added in order to reduce ampholyte interference. Proteins were precipitated by adding 0.1% trichloroacetic acid in the presence of 0.15% deoxycholate. Color was measured at 500 nm in a Wallac 1420 VICTOR3TM Multilabel counter spectrophotometer (PerkinElmer, Wellesley, MA, USA).

### 4.9. Presentation of Data and Statistics

Wells were used as experimental units. The normality of the data was tested using the Shapiro-Wilk test. All data from adipocytes cultivated under standard conditions were analyzed by one-way analysis of variance (ANOVA) followed by the Tukey’s honest significant difference post hoc test to detect differences within cells incubated with different FA. All other data where adipocytes were exposed not only to different FA but also to different culture conditions were analyzed by a two-way ANOVA using the FAs and culture conditions as the effects. Differences were considered statistically significant at *p* < 0.05. Values are shown as means ± standard error of the means (SEM). All statistical analyses were conducted using the software JMP^®^ version 13.1.0 (SAS Institute Inc., Cary, NC, 1989-2007) and GraphPad Prism 6 (La Jolla, USA, www.graphpad.com).

## 5. Conclusions

The oil fraction of today’s commercial diets for Atlantic salmon is based on 70% rapeseed oil and 30% fish oil, compared to traditional diets where the oil fraction was purely based on fish oil [19]. This has resulted in a major reduction in EPA, DHA, and PA and an increase in OA in adipose tissue. However, it is not well-known how this change in FA composition affects adipocyte physiology. Our results, however, show that the EPA, OA, and PA exhibit very different and often opposing effects on central adipocyte functions. The type of dietary FA affects Atlantic salmon adipose tissue metabolism and physiology by modulating the transcript level of relevant genes, modifying the lipolytic activity, and modulating several metabolic processes, such as lipid droplet formation, the leptin system, and mitochondrial dynamics. While EPA reduced the degree of lipid droplet formation in the adipocyte and was a potent inhibitor of the formation of cholesterol esters in the cell, OA resulted in higher accumulation of lipid droplets in the cells, and the formation of cholesterol ester was increased in cells enriched with PA. On the other hand, a cellular enrichment in both PA and EPA increased lipid secretion from the adipocyte when a fasting condition was mimicked. Adipocytes responded to a fasting status by decreasing leptin secretion, following the already described model for most mammalian species. However, when adipocytes were not nutrient-deprived, cells rich in EPA had lower intracellular leptin levels than cells rich in PA. Additionally, the mitochondria reacted to changes in the energy supply and demand. An increase in mitochondrial area was observed after 3 h of mimicking a fasting condition, particularly in cells enriched in OA or EPA. Overall, this in vitro study highlights the role of FA as potential modulators of Atlantic salmon adiposity and demonstrates that single FA possess different metabolic signaling capacities in Atlantic salmon adipocytes. Further studies are required in order to fully understand how changes in the lipid composition of adipocytes affects fish physiology and health, particularly in moments where the fish stop eating, such as during spawning or during an outbreak of certain diseases. In these situations, the capacity to recruit lipids from adipocytes would be essential for reproduction success and health recovery after disease.

## Figures and Tables

**Figure 1 ijms-21-02332-f001:**
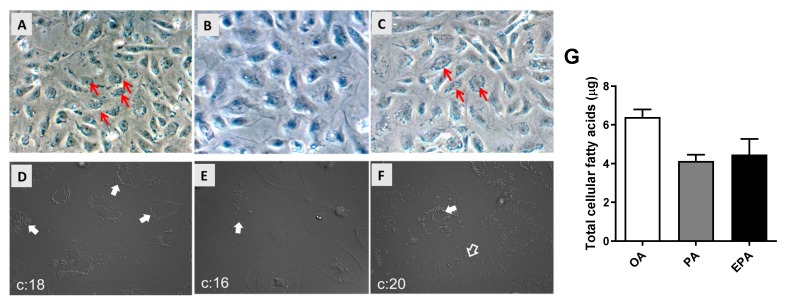
Microscopic observation of lipid droplets (**A**–**F**) and total fatty acid content quantified by gas chromatography (**G**) in mature Atlantic salmon adipocytes. Differentiated cells at day 9 were incubated with oleic, palmitic, or eicosapentaenoic acid for 72 h. Images were taken with a 20× magnification. All arrows, regardless of color, point to areas with a high presence of lipid droplets. Data are shown as mean + SEM (*n* = 3) and analyzed by one-way ANOVA (*p* = 0.0548). OA = oleic acid, PA = palmitic acid, and EPA = eicosapentaenoic acid.

**Figure 2 ijms-21-02332-f002:**
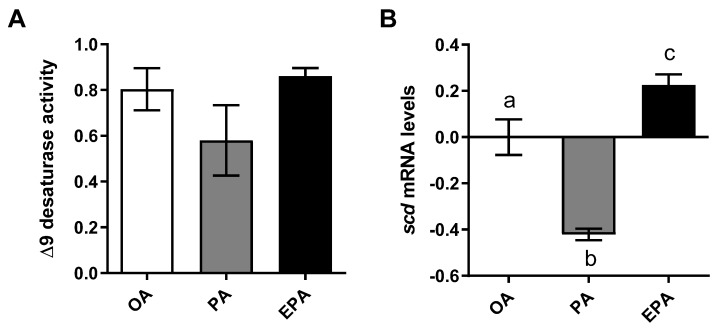
Delta 9 desaturase activity (**A**) and relative changes in transcript levels (**B**) in mature Atlantic salmon adipocytes incubated with oleic, palmitic, or eicosapentaenoic acid for 72 h. Data are shown as mean ± SEM (*n* = 3 for Δ9 desaturase activity, and *n* = 4 for the relative changes in transcript levels) and analyzed by one-way ANOVA followed by Tukey’s honestly significant difference test (*p* = 0.2197 and *p* < 0.0001, respectively). Different letters indicate significant differences between treatments. The activity was calculated based on the 16:0 and 16:1+18:1 produced from [1-^14^C] PA. OA = oleic acid, PA = palmitic acid, and EPA = eicosapentaenoic acid. Relative changes in transcript levels were analyzed with real-time qPCR; data are presented as −ΔΔCt ± SEM, and the OA group was set to zero.

**Figure 3 ijms-21-02332-f003:**
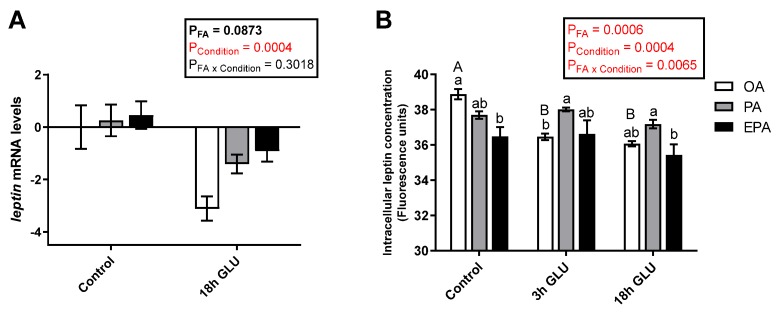
(**A**) Relative changes in transcript levels of leptin in mature Atlantic salmon adipocytes incubated with oleic, palmitic, or eicosapentaenoic acid for 72 h (Control) and serum-deprived in the presence of glucagon thereafter for 18 h (18h GLU). Data are shown as mean ± SEM (*n* = 4). Results are compared by two-way ANOVA using the fatty acid tested (OA, PA, and EPA) and the experimental condition (Control and 18h GLU) as factors. (**B**) Changes in intracellular leptin concentrations in mature Atlantic salmon adipocytes incubated with oleic, palmitic, or eicosapentaenoic acid for 72 h (Control) and serum-deprived in the presence of glucagon thereafter for 3 (3h GLU) and 18 h (18h GLU). Data are shown as mean ± SEM (*n* = 3). Results are compared by two-way ANOVA using the fatty acid tested (OA, PA, and EPA) and the experimental condition (Control, 3h GLU, and 18h GLU) as factors. Lowercase letters indicate significant differences between fatty acids, and capital letters indicate significant differences between conditions (*p* < 0.05; Tukey’s post hoc test). Relative changes in leptin transcript levels were analyzed with real-time qPCR; data are presented as −ΔΔCt ± SEM, and the OA group from the Control condition was set to zero. OA = oleic acid, PA = palmitic acid, and EPA = eicosapentaenoic acid.

**Figure 4 ijms-21-02332-f004:**
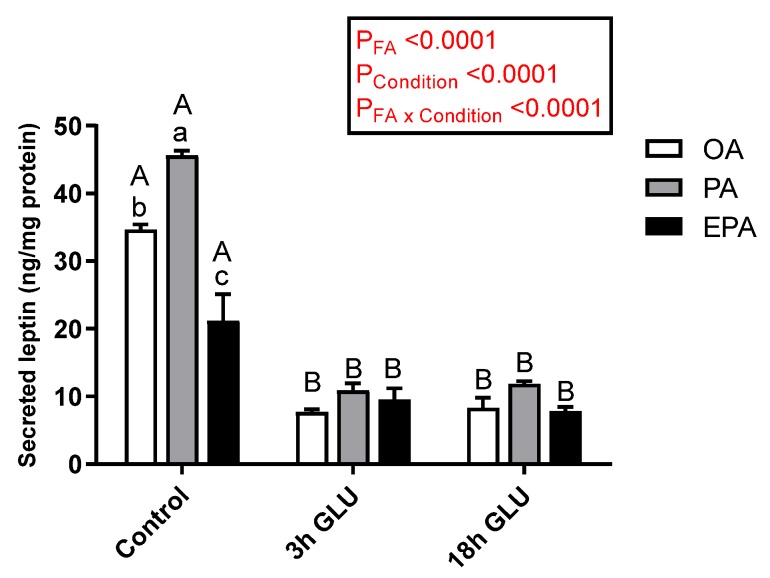
Changes in leptin secretion in the medium from mature Atlantic salmon adipocytes incubated with oleic, palmitic, or eicosapentaenoic acid for 72 h (Control) and serum-deprived in the presence of glucagon thereafter for 3 (3 h GLU) and 18 h (18 h GLU). Data are shown as mean ± SEM (*n* = 3). Results are compared by two-way ANOVA using the fatty acid tested (OA, PA, and EPA) and the experimental condition (Control, 3 h GLU, and 18 h GLU) as factors. Lowercase letters indicate significant differences between fatty acids, and capital letters indicate significant differences between conditions (*p* < 0.05; Tukey’s post hoc test). OA = oleic acid, PA = palmitic acid, and EPA = eicosapentaenoic acid.

**Figure 5 ijms-21-02332-f005:**
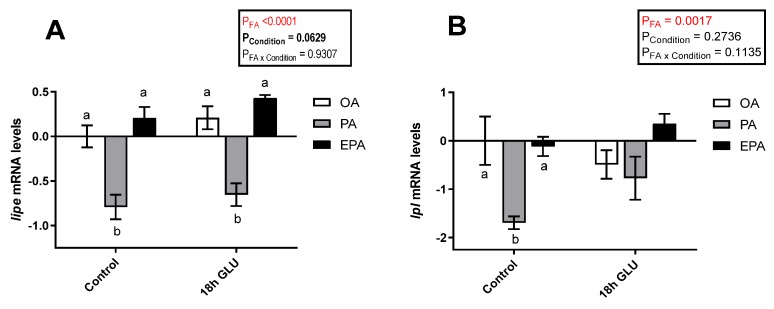
Relative changes in transcript levels of hormone-sensitive lipase (*lipe*) (**A**) and lipoprotein lipase (*lpl*) (**B**) in matured Atlantic salmon adipocytes incubated with oleic, palmitic, or eicosapentaenoic acid for 72 h (Control) and serum-deprived in the presence of glucagon thereafter for 18 h (18 h GLU). Samples (*n* = 4) were analyzed with real-time qPCR; data are presented as −ΔΔCt ± SEM, and the OA group from the Control condition was set to zero. Results are compared by two-way ANOVA using the fatty acid tested (OA, PA, and EPA) and the experimental condition (Control and 18 h GLU) as factors. Lowercase letters indicate significant differences between fatty acids (*p* < 0.05; Tukey’s post hoc test). OA = oleic acid, PA = palmitic acid, and EPA = eicosapentaenoic acid.

**Figure 6 ijms-21-02332-f006:**
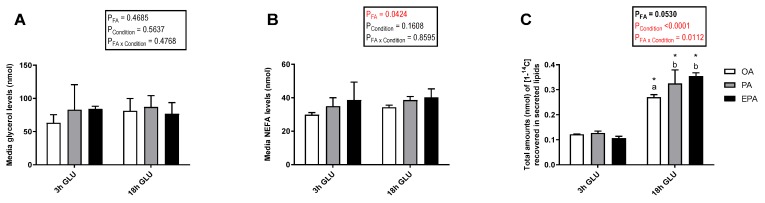
Total glycerol (**A**), non-esterified free fatty acids (NEFA) (**B**), and [1-^14^C] from radiolabelled PA recovered in secreted lipids (**C**) in the media from mature Atlantic salmon adipocytes incubated with oleic, palmitic, or eicosapentaenoic acid for 72 h and serum-deprived in the presence of glucagon thereafter for 3 (3 h GLU) and 18 h (18 h GLU). Data are shown as mean ± SEM (*n* = 4 for glycerol and NEFA data, and *n* = 3 for radiolabeled lipids secreted data). Results are compared by two-way ANOVA using the fatty acid tested (OA, PA, and EPA) and the experimental condition (3 h GLU and 18 h GLU) as factors. Lowercase letters indicate significant differences between fatty acids (*p* < 0.05; Tukey’s post hoc test). Asterisks (*) indicate significant differences between conditions (*p* < 0.05; Sidak’s test). OA = oleic acid, PA = palmitic acid, and EPA = eicosapentaenoic acid.

**Figure 7 ijms-21-02332-f007:**
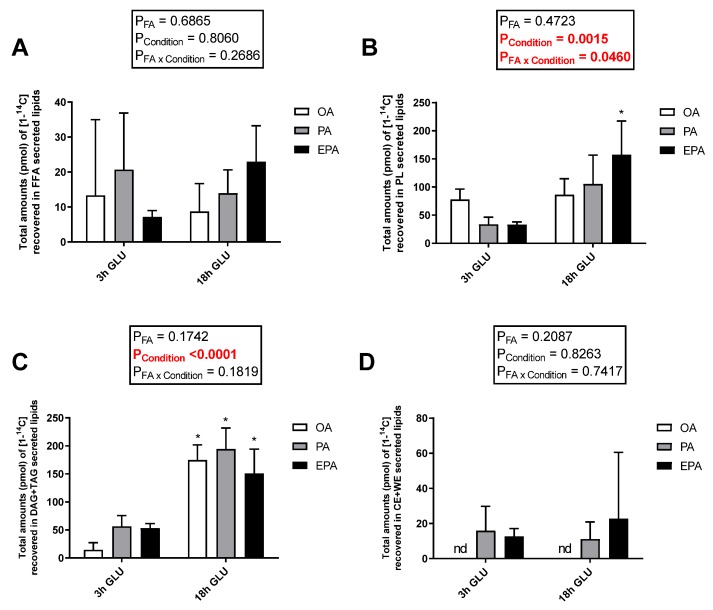
Distribution between free fatty acids (FFA) (**A**), phospholipids (PL) (**B**), mono and diacylglycerol (MDG) and triglycerides (TAG) (**C**), and cholesteryl esters and wax esters (CE + WE) (**D**) produced from [1-^14^C] PA in the media from mature Atlantic salmon adipocytes incubated with oleic, palmitic, or eicosapentaenoic acid for 72 h and serum-deprived in the presence of glucagon thereafter for 3 (3 h GLU) and 18 h (18 h GLU). Data are shown as mean ± SEM (*n* = 3). Results are compared by two-way ANOVA using the fatty acid tested (OA, PA, and EPA) and the experimental condition (3 h GLU and 18 h GLU) as factors. Asterisks (*) indicate significant differences between conditions (*p* < 0.05; Sidak’s test). OA = oleic acid, PA = palmitic acid, and EPA = eicosapentaenoic acid.

**Figure 8 ijms-21-02332-f008:**
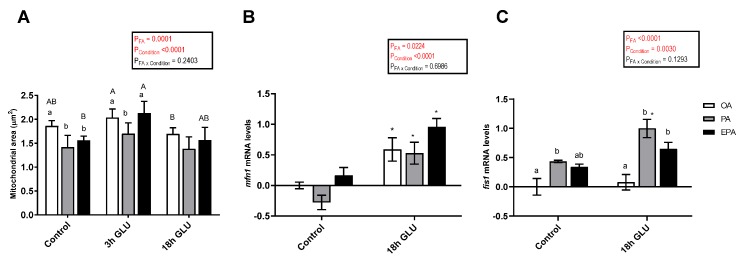
(**A**) Mitochondria area from immunostaining images in mature Atlantic salmon adipocytes incubated with oleic, palmitic, or eicosapentaenoic acid for 72 h (Control) and serum-deprived in the presence of glucagon thereafter for 3 (3 h GLU) and 18 h (18h GLU). Data are shown as mean ± SEM (*n* = 4–6). Results are compared by two-way ANOVA using the fatty acid tested (OA, PA, and EPA) and the experimental condition (Control, 3 h GLU, and 18 h GLU) as factors. Lowercase letters indicate significant differences between fatty acids and capital letters indicate significant differences between conditions (*p* < 0.05; Tukey’s post hoc test). (**B**) Relative changes in transcript levels of *mfn1* and *fis1* (**C**) in mature Atlantic salmon adipocytes incubated with oleic, palmitic, or eicosapentaenoic acid for 72 h (Control) and serum-deprived in the presence of glucagon thereafter for 18 h (18 h GLU). Samples (*n* = 4) were analyzed with real-time qPCR; data are presented as −ΔΔCt ± SEM, and the OA group from the Control condition was set to zero. Results are compared by two-way ANOVA using the fatty acid tested (OA, PA, and EPA) and the experimental condition (Control and 18 h GLU) as factors. Lowercase letters indicate significant differences between fatty acids (*p* < 0.05; Tukey’s post hoc test). Asterisks (*) indicate significant differences between conditions (*p* < 0.05; Sidak’s test). OA = oleic acid, PA = palmitic acid, and EPA = eicosapentaenoic acid.

**Table 1 ijms-21-02332-t001:** Fatty acid composition (% of total) in mature Atlantic salmon adipocytes. Differentiated cells at day 9 were incubated with oleic (OA), palmitic (PA), or eicosapentaenoic acid (EPA) for 72 h (mean ± SEM; *n* = 3).

Fatty Acid	OA	PA	EPA	*p*
16:0	10.0 ± 4.5b	36.4 ± 4.3a	19.0 ± 0.8b	0.0058
Other SFA§	15.5 ± 0.7	17.5 ± 1.8	16.3 ± 3.1	0.8031
16:1 *n*−7	nd	3.8 ± 3.8	1.0 ± 0.4	0.4910
18:1 *n*−7	1.9 ± 0.3	5.0 ± 2.6	2.0 ± 0.1	0.3218
18:1 *n*−9	47.4 ± 6.7a	15.8 ± 1.0b	17.5 ± 0.2b	0.0020
Other MUFA†	8.5 ± 3.4	9.9 ± 5.9	8.4 ± 1.2	0.9593
18:2 *n*−6	3.7 ± 1.0	2.2 ± 0.6	3.8 ± 0.1	0.2617
20:4 *n*−6	4.6 ± 0.6	4.0 ± 0.9	3.3 ± 0.5	0.4239
20:5 *n*−3	ND	2.9 ± 1.3b	18.6 ± 3.1a	0.0010
22:5 *n*−3	2.1 ± 0.3	4.3 ± 2.7	3.5 ± 0.4	0.6279
22:6 *n*-−3	4.0 ± 0.4	2.6 ± 0.7	3.3 ± 0.4	0.2751
Other FA‡	5.1 ± 0.0	0.7 ± 0.0	3.2 ± 0.7	0.2252
Sum identified	97.5 ± 0.8	96.0 ± 1.9	96.8 ± 0.6	0.6994
Sum EPA/DHA	4.0 ± 0.4b	5.6 ± 0.6b	21.9 ± 3.3a	0.0011
Sum N−3	7.8 ± 1.2b	10.1 ± 2.2b	26.4 ± 3.9a	0.0053
Sum N−6	8.2 ± 1.6	6.2 ± 1.2	9.1 ± 0.6	0.2900
Sum N−0	25.5 ± 4.8b	53.9 ± 5.9a	35.3 ± 3.9ab	0.0172

^a,b^ Mean values within a row with unlike superscript letters were significantly different (*p* < 0.05; one-way ANOVA followed by Tukey’s honestly significant difference test). ND, not detectable levels. §Other saturated fatty acids (SFA): including 14:0, 15:0, 18:0, 20:0, and 22:0. †Other monounsaturated fatty acids (MUFA): including 20:1*n*−11, 20:1 *n*−9, and 22:1 *n*−11. ‡Other fatty acids (FA): including 18:3 *n*−3, 20:2 *n*−6, and 20:3 *n*−6.

**Table 2 ijms-21-02332-t002:** Relative distribution of radioactivity from [1-^14^C PA] recovered in different cellular lipid classes in Atlantic salmon differentiated adipocytes incubated with oleic (OA), palmitic (PA), or eicosapentaenoic acid (EPA) for 72 h.

(%)	OA	PA	EPA	*p*
PL	46.17 ± 3.37 ^b^	47.26 ± 7.02 ^b^	82.33 ± 2.71 ^a^	0.0026
FFA	2.63 ± 0.25 ^b^	12.00 ± 2.28 ^a^	3.35 ± 0.97 ^b^	0.0064
TAG	5.84 ± 1.10 ^b^	11.28 ± 1.32 ^ab^	11.93 ± 1.55 ^a^	0.0338
DAG	2.08 ± 0.43	1.9 ± 0.72	2.39 ± 0.67	0.8583
CE	43.29 ± 4.67 ^a^	27.56 ± 4.56 ^b^	ND	0.0367
Total nmol ^1^	5.8 ± 0.35 ^a^	1.8 ± 0.07 ^b^	6.7 ± 0.46 ^a^	0.0001

Data are shown as mean ± SEM (*n* = 3). Different letters indicate significant differences between the experimental groups (*p* < 0.05; one-way ANOVA followed by Tukey’s honestly significant difference test). The recovery values for the OA, PA, and EPA group were 87.27 ± 2.91%, 97.35 ± 3.05%, and 89.51 ± 3.18% of the initial radiolabelled PA, respectively. ND, not detectable levels. ^1^Total nmol: total nmol radiolabeled lipids recovered in the cellular lipid fraction. PL = phospholipids, FFA = free fatty acids, TAG = triglycerides, DAG = diacylglycerol, and CE = cholesteryl esters.

**Table 3 ijms-21-02332-t003:** Atlantic salmon primer sequences used for real-time PCR.

Gene	Accession No.	Direction	Primer Sequence 5′→ 3′
*ef1α*	AF321836	Forward	CACCACCGGCCATCTGATCTACAA
		Reverse	TCAGCAGCCTCCTTCTGAACTTC
*etif3*	DW542195	Forward	CAGGATGTTGTTGCTGGATGGG
		Reverse	ACCCAACTGGGCAGGTCAAGA
*β-actin*	AF012125	Forward	ACATCAAGGAGAAGCTGTGC
		Reverse	GACAACGGAACCTCTCGTTA
*scd*	BT044999	Forward	TGAAATAGTGCTGTCCCGGGCTC
		Reverse	TGGGGAAACCTCTTAGCCACTCCG
*leptin*	FJ830677	Forward	CCAGGCCGCCAGCAGAAACA
		Reverse	GCGCCACTGGACCCACACTC
*lipe*	XM_014201599	Forward	TCCCCAGACGTTTGTGTCAGATGC
		Reverse	GCTTTGGATCCCCCATTAGTTCCTG
*lpl*	BI468076	Forward	TGCTGGTAGCGGAGAAAGACAT
		Reverse	CTGACCACCAGGAAGACACCAT
*mfn1*	BT072406	Forward	AGTGTGTCCAGTCTTCCGCACA
		Reverse	ACAGGCTACAGCACCCAACCTT
*fis1*	BT072691	Forward	CCCCAGGGGGCATCCTGTCTTA
		Reverse	TTGCAGCTGGCCGATCTAGCG

Elongation factor 1A (*ef1α*), eukaryotic translation initiation factor 3 (*etif3*), Δ9 desaturase (*scd*), hormone-sensitive lipase (*lipe*), lipoprotein lipase (*lpl*), mitofusin 1 (*mfn1*), and mitochondrial fission 1 protein (*fis1*).

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
