# Peer review of "Lipid Deposition and Mobilisation in Atlantic Salmon Adipocytes"

_ijms, 2020, doi:10.3390/ijms21072332_

Round 1

Reviewer 1 Report

This manuscript describes the lipolytic response of salmon adipocytes to different fatty acids in cell culture conditions. Finding largely support FA-specific effects on mechanisms regulating the lipolytic response. This type of research is important to understand how dietary lipids regulate nutrient utilization and how varying lipid source (ie, from fish oil to plant oil) affects these mechanisms. While the data are presented clearly and the findings are discussed efficiently, there are some concerns that the authors might consider during the revision process. Of particular concern is the comparison of treatments across time without an appropriate control condition and the interpretation of numerical differences (P>0.05) as a significant treatment response. These concerns are further described below.

Figure 1: The accumulation and size of lipid droplets should be quantified to support statements regarding the abundance of total cellular fatty acids. The complication with this data set is that the FA effect on intracellular fatty acids are not significant at P<0.05 yet the two data sets (microscopy and fatty acid accumulation) are presented with the assumption of treatment effects. With P = 0.054 the reader will likely make their own conclusion that the numerical differences are significant, but quantifying the data from the microscopy will provide valuable support for these conclusions.

Figure 2: Use of the term “influenced” should not be used to describe the numerical difference in treatment effects with a P value of 0.2197. In this case means are considered similar. Was there consideration to add an additional round of cell culture experiments to increase the sample number from n=3 to n=6? To elucidate treatment effects from only 3 biological replicates is difficult unless biological variation is very low and/or treatment effects are large in magnitude. Adding additional samples will increase confidence that treatment effects do occur and avoids the subjectivity of associating responses when P>0.05. In several instances adding a second round of culture experiments likely would have been useful for pulling out treatment effects that approach significance.

One of the major concerns I have with several of the data sets (Figures 3-7) is the comparison of treatment effects over time (Control (0hr) vs 3hr vs 18hr). The culture conditions change between Control and the two later time points (serum deprivation + glucagon) that mimic fasting, although there is no sample that maintains the Control culture conditions throughout this period. Therefore it is not recommended to test for a main effect of condition because technically the effect of fasting is confounded with any effect of time and it is not possible to pull apart these effects. However, the authors are assuming that any effect of condition is due to the fasting signals which I acknowledge is likely to be true but the proper control condition must be carried out through 18 hr to demonstrate this with confidence. The effect of condition (fasting) has only a minor role in the discussion (lines 348-351 and 369-384) and these effects do not contribute significantly to the major conclusions regarding the functional role of individual FAs. It is recommended that the authors avoid referring to the main effect of condition as a fasting effect and avoid drawing any associated conclusions in this manuscript. The alternative is to perform an additional round of studies in which the standard growth media +FAs is maintained as an appropriate control throughout each sampling period.

The authors are very thorough with their description of the biochemical mechanisms affected by the FA of interest. Integration of these findings with current issues regarding aquafeed formulations and their effect on fish metabolism and health could use improvement which would effectively broaden the scope of the significance of this study. Revising the conclusion with this concept in mind would effectively translate the findings to industry relevance.

Minor issues:

Clarification should be provided in the methods on the y-axis of the gene expression regarding what the fold change values are in reference to.

The term “might contribute” and “might differ” in Iines 144 and 145 is arbitrary and should be avoided. Instead, more accurately descriptive terms for the presence or absence of a treatment effect should be used.

Author Response

This manuscript describes the lipolytic response of salmon adipocytes to different fatty acids in cell culture conditions. Finding largely support FA-specific effects on mechanisms regulating the lipolytic response. This type of research is important to understand how dietary lipids regulate nutrient utilization and how varying lipid source (ie, from fish oil to plant oil) affects these mechanisms. While the data are presented clearly and the findings are discussed efficiently, there are some concerns that the authors might consider during the revision process. Of particular concern is the comparison of treatments across time without an appropriate control condition and the interpretation of numerical differences (P>0.05) as a significant treatment response. These concerns are further described below.

Figure 1: The accumulation and size of lipid droplets should be quantified to support statements regarding the abundance of total cellular fatty acids. The complication with this data set is that the FA effect on intracellular fatty acids are not significant at P<0.05 yet the two data sets (microscopy and fatty acid accumulation) are presented with the assumption of treatment effects. With P = 0.054 the reader will likely make their own conclusion that the numerical differences are significant, but quantifying the data from the microscopy will provide valuable support for these conclusions.

As indicated by the reviewer, we agree that quantifying the content of the lipid droplets by staining methods such as oil red O for instance, would have strengthened our visual observations. Unfortunately, this technique was not used in this experiment. We believe that the visual observation together with the quantification of total fatty acids by gas chromatography together provides enough evidence for the effect of the different fatty acids on the adipocytes. Furthermore, our group has previously quantified lipid droplets with oil red O in adipocytes treated with some of the same fatty acids used in the current experiment (i.e. oleic acid and eicosapentaenoic acid) [1]. Those results agreed with our visual observations from the current experiment and we therefore feel confident with our conclusions. The fact that our observations are in line with those from previous studies is also thoroughly discussed in the discussion section (lines 303-311).

Figure 2: Use of the term “influenced” should not be used to describe the numerical difference in treatment effects with a P value of 0.2197. In this case means are considered similar. Was there consideration to add an additional round of cell culture experiments to increase the sample number from n=3 to n=6? To elucidate treatment effects from only 3 biological
replicates is difficult unless biological variation is very low and/or treatment effects are large in magnitude. Adding additional samples will increase confidence that treatment effects do occur and avoids the subjectivity of associating responses when P>0.05. In several instances adding a second round of culture experiments likely would have been useful for pulling out treatment effects that approach significance.

Indeed, increasing the number of replicates would provide a higher statistical power. However, each individual sample in our study represents a pooled cell sample from 5 fish (meaning 15 fish behind each value), which gives quite good strength. This information is now included in materials and method section in order to show that there are in average cells from 15 fish (lines 467-468; 482-483; 497-498). We have reformulated the sentence and removed the term “influenced” to avoid any possible confusion. However, we believe that this result is worth being described. This change can be found from line 113-117: “Despite of lack of significance, cells containing a high endogenous level of PA (PA group) presented a lower numerical Δ9 desaturation capacity than cells containing a lower level of this FA (OA and EPA groups) (ANOVA; P = 0.2197) (Figure 2A).”

One of the major concerns I have with several of the data sets (Figures 3-7) is the comparison of treatment effects over time (Control (0hr) vs 3hr vs 18hr). The culture conditions change between Control and the two later time points (serum deprivation + glucagon) that mimic fasting, although there is no sample that maintains the Control culture conditions throughout this period. Therefore it is not recommended to test for a main effect of condition because technically the effect of fasting is confounded with any effect of time and it is not possible to pull apart these effects. However, the authors are assuming that any effect of condition is due to the fasting signals which I acknowledge is likely to be true but the proper control condition must be carried out through 18 hr to demonstrate this with confidence. The effect of condition (fasting) has only a minor role in the discussion (lines 348-351 and 369-384) and these effects do not contribute significantly to the major conclusions regarding the functional role of individual FAs. It is recommended that the authors avoid referring to the main effect of condition as a fasting effect and avoid drawing any associated conclusions in this manuscript. The alternative is to perform an additional round of studies in which the standard growth media +FAs is maintained as an appropriate control throughout each sampling period.

We appreciate the input from the reviewer. Indeed, it would have been ideal to have a control for each time point that we were evaluating. However, in this experiment we are focusing on the effects of a short term induced fasting condition, and therefore we believe that the control we are using represents the cells under a “fed status”. Based on our previous experience, when adipocytes are in a mature state and are kept for further 18 hours in standard growth medium, they accumulate more lipids with time, that means the opposite response of what is observed during fasting condition when the cells loose lipids, so we thrust that the response observed is in fact a result of fasting. Our research group has been extensively working on the characterization of Atlantic salmon primary adipocytes and based on our experience, when the cells are already mature, we would need a longer period than 18 hours to see significant differences between cells that were incubated in standard growth conditions.

The authors are very thorough with their description of the biochemical mechanisms affected by the FA of interest. Integration of these findings with current issues regarding aquafeed formulations and their effect on fish metabolism and health could use improvement which would effectively broaden the scope of the significance of this study. Revising the conclusion with this concept in mind would effectively translate the findings to industry relevance.

The conclusion has now been revised as suggested by the reviewer taking into consideration the potential impact for the aquaculture industry. The new conclusion can be found in lines 608-632: “The oil fraction of today’s commercial diets for Atlantic salmon is based on 70% rapeseed oil and 30% fish oil, compared to traditional diets where the oil fraction was purely based on fish oil [19]. This has resulted in a major reduction in EPA, DHA, and PA and an increase in OA in adipose tissue. However, it is not well known how this change in FA composition affects adipocyte physiology. Our results, however, show that the EPA, OA, and PA exhibit very different and often opposing effects on central adipocyte functions. The type of dietary FA affects salmon adipose tissue metabolism and physiology by modulating the transcript level of relevant genes, modifying the lipolytic activity, and modulating several metabolic processes, such as lipid droplet formation, the leptin system, and mitochondrial dynamics. While EPA reduced the degree of lipid droplet formation in the adipocyte and was a potent inhibitor of the formation of cholesterol esters in the cell, OA resulted in higher accumulation of lipid droplets in the cells and the formation of cholesterol ester was increased in cells enriched with PA. On the other hand, a cellular enrichment in both PA and EPA increased lipid secretion from the adipocyte when a fasting condition was mimicked. Adipocytes responded to a fasting status by decreasing leptin secretion, following the already described model for most mammalian species. However, when adipocytes were not nutrient deprived, cells rich in EPA had lower intracellular leptin levels than cells rich in PA.
Additionally, the mitochondria reacted to changes in energy supply and demand. An increase in mitochondrial area was observed after 3 hours of mimicking a fasting condition, particularly in cells enriched in OA or EPA. Overall, this in vitro study highlights the role of FA as potential modulators of Atlantic salmon adiposity and demonstrates that single FA possess different metabolic signaling capacities in Atlantic salmon adipocytes. Further studies are required in order to fully understand how the change in lipid composition of adipocytes affects fish physiology and health. Particularly in moments where the fish stop eating, such as during spawning or during an outbreak of certain diseases. In these situations, the capacity to recruit lipids from adipocytes would be essential for reproduction success and health recovery after disease.”

Minor issues:

Clarification should be provided in the methods on the y-axis of the gene expression regarding what the fold change values are in reference to.

This information has been added in the methods section, line 584-585: “Relative quantification of the abundance of transcripts was calculated using the ΔΔCT method and formula -ΔΔCT = -[(Cttarget gene -Ctβ-actin)treatment - (Cttarget gene - Ctβ-actin)control]”. Additionally, more information has been added in the legend of all figures containing transcript abundance data for the sake of clarity.

The term “might contribute” and “might differ” in Iines 144 and 145 is arbitrary and should be avoided. Instead, more accurately descriptive terms for the presence or absence of a treatment effect should be used.

The sentence has been modified to “The type of FA supplemented to the adipocytes contributed to leptin regulation at both transcript and protein levels, but the effect differed across the nutritional/physiological status of the cell.”

References
1. Todorčević, M.; Vegusdal, A.; Gjoen, T.; Sundvold, H.; Torstensen, B. E.; Kjaer, M. A.; Ruyter, B., Changes in fatty acids metabolism during differentiation of Atlantic salmon preadipocytes; effects of n-3 and n-9 fatty acids. Biochim. Biophys. Acta, Mol. Cell Biol. Lipids 2008, 1781, (6-7), 326-35.

Reviewer 2 Report

I had the opportunity to read the international Journal of Molecular Sciences study: Ref N°: ijms-735742 by Marta Bou et al. The study evaluated adipocyte lipid dynamic and leptin production and secretion as well as mitochondrial responses in Atlantic salmon adipocyte pre-enriched with three different fatty acids (FA): palmitic acid (16:0;PA), oleic acid (18:1n-9; OA), and eicosapentaenoic acid (20:5n-3; EPA)under a fed and a fasting condition. The three FA were selected on basis of their relevance on current feeding practices. This interesting in vitro study provides very valuable information on lipolysis and re-esterification of free fatty acids in fish that maybe useful to understand these processes and, for aquaculture industry, during the feed formulations to ensure an optimal degree of fish adiposity. In my opinion, this manuscript deserves to be published, however, some sections need to be improved.

As specific comments, there are:

Line 63-64 Perhaps, a paraphrase should be added to explain the use of these 3 FA in feeding practices and their effects on the final aquaculture product in term of fillet yield and quality.

Line 64-65 another should explain why the choice of Atlantic salmon as the biological model was chosen over other species of fish or terrestrial vertebrates. This should be well explained.

Line 77-79 “The microscopic observations agreed with the quantification of total FA by GC analyses, showing that the OA group contained a higher level of total cellular fatty acids than the EPA and PA groups (ANOVA; P = 0.0548)”, I don't think that's the right interpretation because of the non-significant test. Please improve it.

Similarly, Is it possible for authors to improve the resolution quality of images from 1D to 1F?

Line 84 the results of total cellular fatty acids quantified by gas chromatography are announced in the caption as mean ± SEM but they are presented as mean + SEM on the figure G. Please improve it.

Line 87-88, line 88-90 and line 90-91 In my opinion, the level of significance of the tests used in comparison of the FA composition in mature Atlantic salmon adipocytes should be included in the text for clarity.

Authors use throughout the text the term of “Atlantic salmon” 13 times (see line 16, 44, 64,134,283, 285, 343, 377, 381, 415, 474,477,552). But, why this terminology is not kept for clarity on lines 82,93,112,158,163,180,202,232,243,267,274,290,312,313,319,407,408,410 and 420.

Line 80 Authors should provide scale of the images from 1A to 1F.

Line 103-107 Here again, the one-way ANOVA test performed on the influence of the endogenous FA composition on the Delta 9 desaturase activity (fig 2A) was not significant. I suggest rewording this presentation.

Please also name the x-axis of the figures from 1 to 8 for clarity.

Line 107-110 significant reduction as well as significant higher in the transcript abundance of the gene encoding the Δ9-desaturase enzyme (scd mRNA levels) should be supported by providing precise data of statistical tests.

Line 114 Data are mean ± SEM in fig. 2 B and mean + SEM in the figure 2A. Please improve it.

Line 84, line 113-114 caption of fig. 1 lipid droplets and total cellular FA, line 115 caption of fig.2 and line 136-138 caption of Table 2, for total FA content, FA composition, lipogenesis and incorporation of radiolabelled PA in different cellular lipid classes, the sentences of ”Data are shown as mean ± SEM” and “analysed by one-way ANOVA” appear repeatedly. This should be better presented in the material and method section to help shorten too long captions.

Line 158-169 (fig.3 leptin mRNA and intracellular response), 180-187(fig.4 secreted leptin response), 201-207 (fig.5 transcriptional response), 231-239 (fig.6 & 7 lipolysis)and 267-280 (fig.8 mitochondrial response, the sentences of ”Data are shown as mean ± SEM” and “analysed by one-way ANOVA” appear repeatedly in captions. This should be better presented in the material and method section to help shorten too long captions.

Line 157 What does the asterisk in Figure 3a mean? Its meaning is missing in the legend.

From fig. 2 to fig. 8 I suggest standardizing the labeling of significant difference between the 3 FAs within the same experimental condition (lowercase letters) and the notation between conditions: standard growth (control) and cultured conditions (asterisks or capital letters) for clarity and easy reading of the results.

Line 214-217 “Nevertheless, there was a significant main effect of the FA used during standard culture conditions on the amount of NEFA secreted (P = 0.0424). Thus, adipocytes incubated with EPA secreted a higher amount of NEFA than adipocytes incubated with OA whereas adipocytes incubated with PA had intermediate levels.”, but the fig. 6B does not accurately report the same information. Please report this important information on the fig. 6B.

Line 375-380 responses to FA in fed and fasting conditions may change even within the same fish species depending on experimental design and protocol and sample size as well as on life stage and gender, this demonstrates the importance of a good description of the material and method used. In Line 415-416, detailed information on Atlantic salmon collected if available (number, size: length and weight as mean ± SEM, date and time between collection and dissection). Similarly, line 438-440 or 463-464, origin of FA used should be clearly indicated, how the authors justify the choice of these FA doses in experiments (while line 64-65, it reported content of OA increase and that of PA and EPA decrease in feeding practices) and references should also be provided.

Line 573-587 Authors should also mention how their findings may be useful in feed formulation to ensure an optimal degree of fish adiposity in Atlantic salmon and other fish species or perhaps further studies are still needed.

Line 671 the name of journal “Lipids Health Dis. » is missing

Line 750 the name of journal “Biochim Biophys Acta” is missing

Author Response

I had the opportunity to read the international Journal of Molecular Sciences study: Ref N°: ijms-735742 by Marta Bou et al. The study evaluated adipocyte lipid dynamic and leptin production and secretion as well as mitochondrial responses in Atlantic salmon adipocyte pre-enriched with three different fatty acids (FA): palmitic acid (16:0;PA), oleic acid (18:1n-9; OA), and eicosapentaenoic acid (20:5n-3; EPA)under a fed and a fasting condition. The three FA were selected on basis of their relevance on current feeding practices. This interesting in vitro study provides very valuable information on lipolysis and re-esterification of free fatty acids in fish that maybe useful to understand these processes and, for aquaculture industry, during the feed formulations to ensure an optimal degree of fish adiposity. In my opinion, this manuscript deserves to be published, however, some sections need to be improved.

As specific comments, there are:
Line 63-64 Perhaps, a paraphrase should be added to explain the use of these 3 FA in feeding practices and their effects on the final aquaculture product in term of fillet yield and quality.

More information has been added in the manuscript regarding this issue; lines 65-68: “Changes in dietary FA composition are reflected in the tissues and organs of the fish [21]. Even though these changes might not impact fish growth performance, they affect the nutritional quality of the fish by reducing the levels of the healthy FA EPA and DHA in their fillets [21].”

Line 64-65 another should explain why the choice of Atlantic salmon as the biological model was chosen over other species of fish or terrestrial vertebrates.

This should be well explained. The use of Atlantic salmon as the biological model has been justified now in lines 68-74: “Atlantic salmon has been ranked as the most efficient aquaculture production system [22, 23], being an important source of protein, omega-3 FA, vitamins, and minerals for human consumption. Atlantic salmon are fed high-lipid diets in commercial production [19]. As in mammals, excess of energy is translated into increased abdominal fat deposition. The functions and development of Atlantic salmon white adipose tissue has many similarities with those from terrestrial vertebrates [5, 6, 24], which makes it a suitable and valuable experimental model.”

Line 77-79 “The microscopic observations agreed with the quantification of total FA by GC analyses, showing that the OA group contained a higher level of total cellular fatty acids than the EPA and PA groups (ANOVA; P = 0.0548)”, I don't think that's the right interpretation because of the non-significant test. Please improve it.

We respectfully disagree with the reviewer. We have objectively described our results and we have never included the term “significant” or “significantly higher” in this statement. With the observation of the cells we could see a difference in the amount of lipid droplets formed in the different experimental groups. We are aware that this is a descriptive result, but these observations were in line with the results from the analysis of the total cellular fatty acid content. Even though the P value obtained was 0.0548, it is widely accepted to discuss these results as "statistically marginally significant". However, and in order to avoid any possible confusion, we are only describing those results and not including the term “significant”.

Similarly, Is it possible for authors to improve the resolution quality of images from 1D to 1F?

Unfortunately, this was the best resolution we could obtain using the differential interference contrast in our brightfield microscope.

Line 84 the results of total cellular fatty acids quantified by gas chromatography are announced in the caption as mean ± SEM but they are presented as mean + SEM on the figure G. Please improve it.

This has been corrected accordingly and in line 95 of the manuscript it can be read: “Data are shown as mean + SEM (n = 3)”

Line 87-88, line 88-90 and line 90-91 In my opinion, the level of significance of the tests used in comparison of the FA composition in mature Atlantic salmon adipocytes should be included in the text for clarity.

The P values have been added to the text as suggested by the reviewer and can be found in lines 98, 100, and 102.

Authors use throughout the text the term of “Atlantic salmon” 13 times (see line 16, 44, 64,134,283, 285, 343, 377, 381, 415, 474,477,552). But, why this terminology is not kept for clarity on lines 82,93,112,158,163,180,202,232,243,267,274,290,312,313,319,407,408,410 and 420.

For the sake of clarity, the term “Atlantic salmon” has been added in all the places indicated by the reviewer.

Line 80 Authors should provide scale of the images from 1A to 1F.

These pictures were taken with a camera coupled to the brightfield microscope and therefore we could not add the scale. However, we have included now in the manuscript (lines 92-93) the magnification used to take the pictures.

Line 103-107 Here again, the one-way ANOVA test performed on the influence of the endogenous FA composition on the Delta 9 desaturase activity (fig 2A) was not significant. I suggest rewording this presentation.

We have reformulated the sentence to avoid any possible confusion. However, we believe that this result is worth being described. This change can be found from line 113-117: “Despite of lack of significance, cells containing a high endogenous level of PA (PA group) presented a lower numerical Δ9 desaturation capacity than cells containing a lower level of this FA (OA and EPA groups) (ANOVA; P = 0.2197) (Figure 2A).”

Please also name the x-axis of the figures from 1 to 8 for clarity.

We believe that the figures are properly labeled and that all the information has been provided to make it understandable for the reader.

Line 107-110 significant reduction as well as significant higher in the transcript abundance of the gene encoding the Δ9-desaturase enzyme (scd mRNA levels) should be supported by providing precise data of statistical tests.

The P value supporting these results has been added (lines 119 and 120) as well as the information about the post hoc test done in the legend of Figure 2 (line 126).

Line 114 Data are mean ± SEM in fig. 2 B and mean + SEM in the figure 2A. Please improve it.

Figure 2A has been modified so that it represents the mean ± SEM.

Line 84, line 113-114 caption of fig. 1 lipid droplets and total cellular FA, line 115 caption of fig.2 and line 136-138 caption of Table 2, for total FA content, FA composition, lipogenesis and incorporation of radiolabelled PA in different cellular lipid classes, the sentences of ”Data are shown as mean ± SEM” and “analysed by one-way ANOVA” appear repeatedly. This should be better presented in the material and method section to help shorten too long captions.

We respectfully disagree with the reviewer since it is a requirement that all figure legends should contain enough information to stand alone without the text from the manuscript.

Line 158-169 (fig.3 leptin mRNA and intracellular response), 180-187(fig.4 secreted leptin response), 201-207 (fig.5 transcriptional response), 231-239 (fig.6 & 7 lipolysis)and 267-280 (fig.8 mitochondrial response, the sentences of ”Data are shown as mean ± SEM” and “analysed by one-way ANOVA” appear repeatedly in captions. This should be better presented in the material and method section to help shorten too long captions.

We respectfully disagree with the reviewer since it is a requirement that all figure legends should contain enough information to stand alone without the text from the manuscript.

Line 157 What does the asterisk in Figure 3a mean? Its meaning is missing in the legend.

This mistake has been corrected now and the asterisk has been removed from the figure.

From fig. 2 to fig. 8 I suggest standardizing the labeling of significant difference between the 3 FAs within the same experimental condition (lowercase letters) and the notation between conditions: standard growth (control) and cultured conditions (asterisks or capital letters) for clarity and easy reading of the results.

The labeling of the significant differences is already standardized and it follows the criteria described by the reviewer. Lowercase letters indicate significant differences between fatty acids within the same experimental condition, and capital letters indicate significant differences between conditions when 3 conditions are included (Control, 3h GLU, and 18h GLU). Asterisks indicate significant differences between conditions when only 2 conditions are tested (Control and 18h GLU, or 3h GLU and 18h GLU).

Line 214-217 “Nevertheless, there was a significant main effect of the FA used during standard culture conditions on the amount of NEFA secreted (P = 0.0424). Thus, adipocytes incubated with EPA secreted a higher amount of NEFA than adipocytes incubated with OA whereas adipocytes incubated with PA had intermediate levels.”, but the fig. 6B does not accurately report the same information. Please report this important information on the fig. 6B.

Figure 6B shows that there is an effect of the FA the cells were cultured in, with a P value of 0.0424. As it is stated in the figure legend, the statistical test run is a two-way ANOVA. This means that there is an overall effect of the FA used during standard culture conditions on the amount of NEFA secreted. The aim of ANOVA is to detect whether a factor, in this case the FA used, has a significant effect on a dependent variable (secreted NEFA) globally. We run a post hoc test to detect where the differences were. However, it failed to detect it. This is not uncommon at all, since post hoc tests involve p-value corrections depending on the amount of multiple comparisons needed, making the test “stricter”.

Line 375-380 responses to FA in fed and fasting conditions may change even within the same fish species depending on experimental design and protocol and sample size as well as on life stage and gender, this demonstrates the importance of a good description of the material and method used. In Line 415-416, detailed information on Atlantic salmon collected if available (number, size: length and weight as mean ± SEM, date and time between collection and dissection). Similarly, line 438-440 or 463-464, origin of FA used should be clearly indicated, how the authors justify the choice of these FA doses in experiments (while line 64-65, it reported content of OA increase and that of PA and EPA decrease in feeding practices) and references should also be provided.

We have extended the information about the fish material used in the experiment; lines 435-439: “Atlantic salmon with a mean body weight of 3.46 kg (SEM 0.58) kg and a mean body length of 617 (SEM 3) cm were obtained from Nofima Research station at Averøy, Norway. Fifteen fish were randomly selected and, transported from sea cages to land tanks with recirculating water. One at the time, the fish were anaesthetized with metacain (MS-222; 0.08 g/l) and killed by a sharp blow to the head.”
The information regarding the origin and doses of the FA used can be found in lines 470-474: “The radiolabeled PA was obtained from American Radiolabeled Chemicals, Inc. (St. Louis, MO, USA) and the non-radiolabeled FAs were all supplied from Sigma-Aldrich (St. Louis, MO, USA). The doses were selected based on previous in vitro studies performed on adipocytes [64, 65] and based on extensive experience in our laboratory from previous studies.”

Line 573-587 Authors should also mention how their findings may be useful in feed formulation to ensure an optimal degree of fish adiposity in Atlantic salmon and other fish species or perhaps further studies are still needed.

The conclusion has now been revised as suggested by the reviewer taking into consideration the potential impact for the aquaculture industry together with the need for further studies. The new conclusion can be found in lines 608-632: “The oil fraction of today’s commercial diets for Atlantic salmon is based on 70% rapeseed oil and 30% fish oil, compared to traditional diets where the oil fraction was purely based on fish oil [19]. This has resulted in a major reduction in EPA, DHA, and PA and an increase in OA in adipose tissue. However, it is not well known how this change in FA composition affects adipocyte physiology. Our results, however, show that the EPA, OA, and PA exhibit very different and often opposing effects on central adipocyte functions. The type of dietary FA affects salmon adipose tissue metabolism and physiology by modulating the transcript level of relevant genes, modifying the lipolytic activity, and modulating several metabolic processes, such as lipid droplet formation, the leptin system, and mitochondrial dynamics. While EPA reduced the degree of lipid droplet formation in the adipocyte and was a potent inhibitor of the formation of cholesterol esters in the cell, OA resulted in higher accumulation of lipid droplets in the cells and the formation of cholesterol ester was increased in cells enriched with PA. On the other hand, a cellular enrichment in both PA and EPA increased lipid secretion from the adipocyte when a fasting condition was mimicked. Adipocytes responded to a fasting status by decreasing leptin secretion, following the already described model for most mammalian species. However, when adipocytes were not nutrient deprived, cells rich in EPA had lower intracellular leptin levels than cells rich in PA. Additionally, the mitochondria reacted to changes in energy supply and demand. An increase in mitochondrial area was observed after 3 hours of mimicking a fasting condition, particularly in cells enriched in OA or EPA. Overall, this in vitro study highlights the role of FA as potential modulators of Atlantic salmon adiposity and demonstrates that single FA possess different metabolic signaling capacities in Atlantic salmon adipocytes. Further studies are required in order to fully understand how the change in lipid composition of adipocytes affects fish physiology and health. Particularly in moments where the fish stop eating, such as during spawning or during an outbreak of certain diseases. In these situations, the capacity to recruit lipids from adipocytes would be essential for reproduction success and health recovery after disease.”

Line 671 the name of journal “Lipids Health Dis. » is missing
This has been fixed in the reference 31 (line 730)

Line 750 the name of journal “Biochim Biophys Acta” is missing
This has been fixed in the reference 62 (line 811)

Reviewer 3 Report

The present work aimed to elucidate how Atlantic salmon adipocytes pre-enriched with palmitic, oleic, or eicosapentaenoic acid, respond to a fasting condition mimicked by nutrient deprivation and glucagon.

The study is well structured; however, I suggest some minor corrections before publication:

  • Please, move M&M section before results
  • Please, specify MS-222 concentration (line 416)
  • Please, provide reference for the National Guidelines for Animal Care and Welfare of the Norwegian Ministry of Research (line 420)
  • Please provide authorization number released by ethical committee
  • 7 or 8, Please, specify the correct number
  • lines 430 and 512 pay attention to superscript and subscript
  • Statistical analysis (line 563): Did you perform a normality test? If yes, what test did you use? On the contrary, if data are not normally distributed, an appropriate non-parametric test should be used (e.g. Kruskal Wallis test).
  • Please specify what SEM means (line 569).
  • Please, check the references layout

Author Response

The present work aimed to elucidate how Atlantic salmon adipocytes pre-enriched with palmitic, oleic, or eicosapentaenoic acid, respond to a fasting condition mimicked by nutrient deprivation and glucagon.
The study is well structured; however, I suggest some minor corrections before publication:
Please, move M&M section before results

We are following the guidelines from the journal and therefore cannot move the M&M section before results.

Please, specify MS-222 concentration (line 416)

This information has been added now in the manuscript, line 438-439: “One at the time, the fish were anaesthetized with metacain (MS-222; 0.08 g/l) and killed by a sharp blow to the head.”

Please, provide reference for the National Guidelines for Animal Care and Welfare of the Norwegian Ministry of Research (line 420)
Please provide authorization number released by ethical committee

We have added now that information in the manuscript; lines 441-445: “The experiment was conducted according to the National Guidelines for Animal Care and Welfare of the Norwegian Ministry of Research (FOR-2015-06-18-761) and classified as not requiring a specific license (§2-f, corresponding to Directive 2010/63/EU Article 1, section 5f), since the experimental treatments were not expected to cause any distress or discomfort for the fish, being that the fish was dead prior to adipose tissue dissection.”

7 or 8, Please, specify the correct number

We have checked and specified the correct number, which can be found in line 447: “Briefly, the dissected adipose tissue from 15 fish was washed with Leibowitz-15…”

lines 430 and 512 pay attention to superscript and subscript
These typos have been corrected (lines 456 and 545).

Statistical analysis (line 563): Did you perform a normality test? If yes, what test did you use? On the contrary, if data are not normally distributed, an appropriate non-parametric test should be used (e.g. Kruskal Wallis test).

We have included now that information in the manuscript; line 598-599: “The normality of the data was tested using the Shapiro-Wilk test.”

Please specify what SEM means (line 569).
This clarification has been done and can be found in line 604: “Values are shown as means ± standard error of the mean (SEM).”
Please, check the references layout

We have checked the references and several changes have been made to comply with the requirements of the journal.

Round 2

Reviewer 1 Report

The authors have adequately addressed my concerns.

Reviewer 2 Report

I am writing with regard to the revised version of the international Journal of Molecular Sciences Manuscript Ref N°: ijms-735742 by Marta Bou et al. I am generally satisfied with the amendments made by the authors.

I think now the manuscript is clearer and I congratulate the authors for the effort done in revising the paper.